# Fc Receptor Variants and Disease: A Crucial Factor to Consider in the Antibody Therapeutics in Clinic

**DOI:** 10.3390/ijms22179489

**Published:** 2021-08-31

**Authors:** Jin Kim, Ji Young Lee, Han Gil Kim, Min Woo Kwak, Tae Hyun Kang

**Affiliations:** 1Department of Interdisciplinary Program for Bio-Health Convergence, Kookmin University, Seoul 02707, Korea; kimjin2094@kookmin.ac.kr; 2Department of Chemistry, Kookmin University, Seoul 02707, Korea; ljy@kookmin.ac.kr; 3Department of Biopharmaceutical Chemistry, Kookmin University, Seoul 02707, Korea; gks954@kookmin.ac.kr (H.G.K.); mwoo@kookmin.ac.kr (M.W.K.)

**Keywords:** FcRs, genetic variants, immunological disorders, infectious diseases, cancer, antibody engineering

## Abstract

The fragment crystallizable (Fc) domain of antibodies is responsible for their protective function and long-lasting serum half-life via Fc-mediated effector function, transcytosis, and recycling through its interaction with Fc receptors (FcRs) expressed on various immune leukocytes, epithelial, and endothelial cells. Therefore, the Fc–FcRs interaction is a control point of both endogenous and therapeutic antibody function. There are a number of reported genetic variants of FcRs, which include polymorphisms in (i) extracellular domain of FcRs, which change their affinities to Fc domain of antibodies; (ii) both cytoplasmic and intracellular domain, which alters the extent of signal transduction; and (iii) the promoter region of the FcRs gene, which affects the expression level of FcRs, thus being associated with the pathogenesis of disease indications. In this review, we firstly describe the correlation between the genetic variants of FcRs and immunological disorders by individual differences in the extent of FcRs-mediated regulations. Secondly, we discuss the influence of the genetic variants of FcRs on the susceptibility to infectious diseases or cancer in the perspective of FcRs-induced effector functions. Overall, we concluded that the genetic variants of FcRs are one of the key elements in the design of antibody therapeutics due to their variety of clinical outcomes among individuals.

## 1. Introduction

In the top 10 best-selling drugs in 2020, a total of 5 were monoclonal antibodies (mAbs), including the first and second rank drugs [1]. In addition, 57.3% of the expected top 10 drug sales in 2021 were therapeutic mAbs, and this proportion is predicted to rise even higher [2]. By May 2021, the Food and Drug Administration (FDA) authorized 102 therapeutic mAbs. As for disease indications, treatments of therapeutic mAbs were carried out mainly in cancers (43.1%), autoimmune diseases (25.5%), and infectious diseases (6.9%) [3]. The major reasons why the market of therapeutic antibodies is growing so rapidly are due to their high specificity, low toxicity, and long serum half-life, similar to their biological properties in our body [4,5,6,7].

Antibodies play protective roles by specific binding to pathogens, infected cells, and tumor cells via antibody-mediated functions, including neutralization [8,9], agglutination [10,11], antibody-dependent cellular cytotoxicity (ADCC) [12,13,14], antibody-dependent cellular phagocytosis (ADCP) [15,16], and complement-mediated cytotoxicity (CDC) [17,18]. These functions stem from the cooperation of the fragment antigen-binding (Fab) that recognizes antigens in a specific manner and the Fc that triggers the effector functions by binding to the FcRs [19]. In addition to the major function of antibodies as pivotal parts in humoral immunity, the interaction between the Fc-portion of antibodies and FcRs can effectively cooperate with FcRs-expressed cellular immunity.

FcRs-mediated antibody effector function, transcytosis, and recycling are initialized by the Fc–FcRs interactions. These events are sophisticated outcomes, being harmonized with the following three features. The first is the inherency of FcRs. The existence of various types of FcRs on the surface of immune leukocytes results in intrinsic affinities and specificity to the Fc domain of antibodies, leading to diverse antibody-mediated immune reactions. FcRs are classified into five types (FcαRs, FcδRs, FcγRs, FcεRs, FcμRs), interacting with the isotypes of immunoglobulins (Igs) [20,21]. Either FcγRs or FcεRs are further categorized into FcγRI, FcγRIIa, FcγRIIb, FcγRIIc, FcγRIIIa, and FcγRIIIb, or FcεRI and FcεRII, respectively [22]. Fcα/μR is a dual binding receptor protein to both IgA and IgM. Despite its unknown function, it is clear that Fcα/μR is related to both positive and negative immune responses, based on recent research [23]. While FcγRs, FcεRs, and FcαRs regulate immune regulation, the neonatal Fc receptor (FcRn) interacts with IgGs and enables them to be recycled in serum [24]. In addition, polymeric immunoglobulin receptor (pIgR) interacts with both dimeric IgA and pentameric IgM and exerts them to be transported from the basolateral surface of the epithelium to the apical side [24,25]. The second characteristic which contributes to immune effector functions is the distribution of FcRs on various cell types. FcRs are expressed on the membranes of not only immune cells, but also epithelial and endothelial cells (Table 1), and their expression profiles are variable in response to surrounding cytokines. The last hallmark is the diversity of FcRs ligands that are distinctive isotypes and subclasses of Igs. Igs are divided into five isotypes (IgA, IgD, IgE, IgG, and IgM) and six further subclasses (IgA1, IgA2, IgG1, IgG2, IgG3, and IgG4) [26,27,28]. Each of them has its own unique structural and functional characteristics, thus triggering various immune responses. Furthermore, B-cells can alter their production of Ig isotypes, regardless of antigen-specificity, through the unique process called immunoglobulin isotype switching, which is caused by recombination of the immunoglobulin heavy chain (IGH) gene. Because each Ig isotype has an intrinsic affinity to FcRs, isotype switching also remodels the recruitment of effector cells, regulated by cytokines with regards to their functional requirements [29,30].

When the immune effector functions are excessive by the cooperative and synergistic effects of the three features, immune self-attack arises and can lead to autoimmune, inflammatory, or allergic diseases by the unbalanced activation of FcRs [21,22,31]. Consequently, these abnormal immune reactions by either autoantibodies or autoreactive lymphocytes target self-antigens with hypersensitivity to harmless environmental antigens, respectively [32,33]. Conversely, if the immune effector functions are insufficient to meet the activation threshold, the susceptibility to pathogens is increased by the non-induction of effector functions, leading to less clearances of immune complexes (ICs) [26,34]. Likewise, immunosurveillance against precancerous or cancerous cells can be attenuated [35,36].

In this review, we discuss (i) the correlations between genetic diversity of FcRs among individuals and pathogenesis of immunological disorders and (ii) the effect of FcRs genetic variants on susceptibility to infectious diseases and cancer. With regards to diverse FcRs variants, antibody engineering strategies for improving therapeutic efficacy are also reviewed.

## 2. Immune Disorders and Fc Receptor Genetic Variants

The genetic variety of FcRs can affect individual FcRs-Ig affinities, the strength of FcRs’ intracellular signaling, and their expression levels, which can lead to the distinctive immunological outcomes. In other words, FcRs’ polymorphisms are closely related to a variety of overactive immune responses such as autoimmunity, inflammation, and allergic reactions. This section describes three different types of FcR variants, which are FcγRIIb, FcεRI, and FcαRI, and their consequent immune activation mechanisms.

### 2.1. FcγRIIb Genetic Variants: Insufficient Inhibitory Function of FcγRIIb Cause Autoimmune Diseases

FcγRIIb, the sole inhibitory receptor of FcγRs known so far, serves as a negative regulator of humoral immunity via the inhibition of B-cell activation or antigen phagocytosis of dendritic cells (DCs). FcγRIIb is expressed on various immune cells such as DCs and macrophages (Table 1), and notably, it is the only FcγR expressed on B-cells and plasma cells as well [21,22,37]. Activated signaling pathways of immune cells are suppressed via the dephosphorylation of activation motifs of B-cell receptors (BCRs) or T-cell receptors (TCRs) through the co-engagement of immunoreceptor tyrosine-based inhibitory motif (ITIM) in the cytoplasmic tail of FcγRIIb. The IgG binding ratio of activating FcγRs to FcγRIIb (A/I ratio) on the immune cells determines the activation threshold, and this can be altered by the extent of FcγRIIb expression [38,39]. In clinical perspectives, FcγRIIb is reported to be involved in the pathogenesis of autoimmune diseases [40,41,42,43]. For example, one of the main hallmarks of autoimmune diseases such as systemic lupus erythematosus (SLE) is the overproduction of autoantibodies due to the defect in the downregulation of the humoral immune response affected by FcγRIIb-dependent inhibitory mechanisms [37,44].

Single nucleotide polymorphisms (SNPs) in the promoter region of the *FCGRIIB* gene are reported to change the affinity to transcription factors and result in expression differences of FcγRIIb on the immune cell surface (Figure 1) [37,39,45]. Three SNPs in the promoter are identified: −386G/C, −343G/C, and −120T/A promoter polymorphisms in the relative positions from the transcription start site (TSS: Position +1) (Figure 1 and Table 2) [46,47]. Interestingly, because the −386C allele mostly accompanies the −120A allele, −386C allele and −120A alleles may form a haplotype [46]. Each promoter with −386C−120A haplotype and −343G allele showed a higher transcription strength of promoter activity with the higher FcγRIIb expression level than that of −386G−120T and −343C allele, respectively (Figure 1). This may be the reason why an insufficient FcγRIIb expression affected by the SNPs is associated with autoimmune diseases (Table 2). The homozygous −343C/C genotype was observed at a higher frequency in SLE patients (7.9%) than in healthy controls (0.8%) [47]. In addition, SLE patients reported decreased expression of FcγRIIb in both memory B cells and DCs [48,49]. Based on these results, immunological disorders can be caused by decreased FcγRIIb expression due to the genetic variations in the promoter region of the *FCGRIIB* gene.

The aforementioned research indicates that reinforcing the affinity of FcγRIIb with engineered Fc might have an advantage in the treatment of autoimmune diseases by the suppression of the activating signal of abnormal immune responses via the FcγRIIb-dependent inhibitory mechanisms [37,48,49,50,51,52,53,54,55,56,57]. For example, Szymkowski and his colleagues constructed an anti-CD19 antibody, XmAb5871, with engineered Fc that exhibits a >400-fold increased FcγRIIb affinity, relative to native IgG1 Fc. XmAb5871 suppressed B cell proliferation and antigen-presenting cell (APC) function by overcoming the FcγRIIb dysfunction of both SLE and rheumatoid arthritis (RA) B cells. Moreover, in both cases, the abnormal production of antibodies was decreased and an improved survival rate was observed after administration of XmAb5871 in both SLE and RA peripheral blood mononuclear cell (PBMC)-engrafted mice [58,59].

### 2.2. FcεRI Genetic Variants: Hyperactive FcεRI Results in Allergic Diseases

FcεRI is one of the high-affinity receptors that have been studied as a critical component in allergic responses. FcεRI is an Fc receptor for IgE isotype antibodies, and the cross-linking of FcεRI by the binding of IgE activates mast cells to secrete allergic mediators such as histamine, leukotrienes, and a number of cytokines and chemokines to induce allergic responses. The FcεRI is observed on the surface of not only mast cells but also epidermal Langerhans cells, eosinophils, basophils, and even antigen-presenting cells such as DCs and monocytes (Table 1) [60,61,62,63].

The heterotetrameric FcεRI is composed of α, β, and two γ subunits linked by disulfide bonds. Each subunit contributes to the function of the FcεRI, and they are reported to have several SNPs in their promoter regions, which affect allergic responses (Figure 1). As for the α subunit, −66T/C and −344T/C alleles have been identified as the promoters of the *FCERIA* gene (Figure 1 and Table 2). The higher strength of promoter activity due to the high affinity of transcription factors to the FcεRI α subunit gene was detected in the −66T and −344T alleles, compared to that of the −66C and −344C alleles, respectively (Figure 1) [64,65]. On one hand, people with a heterozygous −66T/C genotype are generally healthy, while those with a homozygous −66T/T genotype tend to hold allergic diseases, which indicates that people who have −66C allele presents resistance to allergic diseases [65]. On the other hand, −344T allele carriers expressed FcεRI on mast cells, resulting in a high degree of histamine release to provoke IgE-mediated allergic responses [66] and elevation of total serum IgE level [67,68]. As for now, the detailed relationship between the elevated IgE level and SNPs in the promoter is still unknown. In another report, among aspirin-intolerant chronic urticaria (AICU) and aspirin-intolerant asthma (AIA) patients, those who have −344T allele not only indicated a higher atopy rate (among AIA patients, those who had −344T allele were 59.46%; among AICU patients, those who had −344T allele were 84.1%) but also total serum IgE concentrations compared to those with the homozygous −344C/C genotype [66,69]. Collectively, the prevalence of −344T allele with relatively high affinity to transcription factor to the other −344C allele in both AICU and AIA patients indicates that SNPs at the promoter of the *FCERIA* gene plays a significant role in the development of allergic diseases.

In the β subunit, three SNPs have been reported: the promoter region (−109T/C), intracellular domain (G237E), and transmembrane domain (L181I and L183V) of the *FCERIB* gene (Figure 1 and Figure 2, and Table 2). Firstly, individuals with the −109T allele showed a higher expression level of the FcεRI β subunit than those with −109C allele (Figure 1), and also asthma patients with homozygous −109T/T genotype showed significantly higher total IgE level, compared to those with −109T/C or −109C/C genotype [70,71,72]. Secondly, the G237E polymorphism also affects allergic responses which can be explained through the following two hypotheses: (1) G237E polymorphism is placed in the intracellular tail of β subunit close to an immunoreceptor tyrosine-based activation motif (ITAM) (Figure 2). The substitution of glutamic acid to glycine residue introduces a critical hydrophobicity adjacent to ITAM in the C-terminus of the β subunit, which can affect the signaling for FcεRI function [73]. (2) G237E polymorphism, which is close to an alternative splicing site, may also affect expression level of the receptor on the cell surface. Kinet and his colleagues identified a new transcript of the β subunit that contains an in-frame stop codon generated by an alternative splicing of the FcεRI β transcript, which would be translated into a truncated β subunit [74]. Compared to the full-length β subunit, the truncated one is significantly less expressed and therefore the G237E polymorphism might affect the expression level of the β chain, which can lead to the diverse susceptibility of allergic responses among individuals [74,75,76]. Lastly, L181I and L183V polymorphism is located in the transmembrane domain (Figure 2). Detailed mechanisms have not yet been fully studied; however, all the infants who have maternally inherited the FcεRI-L181 allele revealed a higher incidence of atopy than those with the I181 and homozygous FcεRI–L/L181 and L/L183 genotypes showed a higher prevalence in asthma, compared with the counterpart genotype [77,78,79]. In the γ subunit, −237A/G polymorphism on the promoter region in the *FCERIG* gene has been identified (Figure 1). The function of the γ subunit is associated with the signal-transduction pathway of FcεRI. For example, the ITAM of the γ subunit is pivotal for FcεRI-mediated mast cell activation. Individuals with −237A allele showed higher expression of the γ chain than those with −237G allele, which subsequently upregulates the mast cell activation more easily and release inflammatory mediator and cytokines (Figure 1) [66,79,80]. AIA patients with the −237A/A genotype are reported to exhibit higher total serum IgE level than those with the A/G, G/G genotype (IgE concentration of AIA patients with the −237A/A genotype were Log2.25 ± 0.57 IU/mL), which indicates that the −237A/G polymorphism plays a significant role in the development of AIA [69].

Based on these results, preventing the cross-linking of IgE with the FcεRI can be one of the attractive treatments for IgE-mediated allergic diseases. Indeed, omalizumab (Xolair^®^) which is an FDA and European Medicines Agency (EMA)-approved recombinant humanized anti-IgE mAbs that neutralize IgE by blocking IgE–FcεRI interaction [81]. Of note, omalizumab was also approved for self-injection across all the approved allergic indications as of April 2021 [82].

### 2.3. FcαRI Genetic Variants: Upregulated FcαRI Leads to Autoimmune Diseases

FcαRI is expressed on neutrophils, eosinophils, monocytes/macrophages, DCs, and Kupffer cells (Table 1) [83]. This receptor is specific to IgA and can bind to both human IgA1 and IgA2 subclasses [84,85]. Among three of the FcαRI polymorphisms, two exist in the promoter region (−114C/T and +56C/T on the promoter in the *FCAR* gene, Figure 1) and the other in the cytoplasmic domain (G248S, Figure 2), each of which causes diverse immunological impact (Table 2). 

Two polymorphisms exist in the FcαRI promoter and have been reported to be associated with the etiology of IgA nephropathy (IgAN) [86]. IgAN is the most common disease of primary glomerulonephritis (GN) by the accumulation of unglycosylated IgA1-FcαRI complexes in the glomerular mesangium [87,88,89]. −114C and +56C alleles were more frequent in IgAN patients than in other GN patients and healthy adults [86]. The *FCAR* gene promoter with these two alleles has a significantly higher promoter activity than those with −114T and +56T alleles (Figure 1) [90], which upregulate the expression of soluble FcαRI in the serum of IgAN patients, consequently leading to development into IgAN via the increased deposition of the IgA1–FcαRI complex in the mesangium [91,92]. Consistent with these results, removing soluble FcαRI from the serum of transgenic mice expressing human FcαRI in macrophages/monocytes can eliminate effects such as massive mesangial IgA deposition, glomerular, and interstitial macrophage infiltration, mesangial matrix expansion, hematuria, and mild proteinuria, which are the initial symptoms of IgAN [93,94,95].

G248S polymorphism in the cytoplasmic domain of FcαRI affects intracellular signals, thereby leading to different biological outcomes (Figure 2) [96]. FcαRI-S248 allele requires FcαRI γ-chain to mediate cytokine production, whereas FcαRI-G248 allele can mediate pro-inflammatory cytokine production without the γ-chain binding. Genotype analysis confirmed that the G248 allele is significantly enriched in SLE patients (36% for African American, 19% for Caucasian), compared to control populations (26% for African American, 14% for Caucasian). Therefore, the G248 allele may be considered as a candidate risk factor in autoimmune diseases such as SLE [97].

Combined with these studies, the therapeutic direction of IgAN and SLE can be offered with a method to target FcαRI or IgA, which can neutralize soluble FcαRI or unglycosylated IgA1 to interrupt the IgA–FcαRI interaction.

## 3. Therapeutic Efficacy of Antibodies and Fc Receptor Genetic Variants

The genetic variety of FcRs induces different effector functions by both endogenous and therapeutic antibodies, conferring tolerance to biological invasions and malignant cells. Thus, the genetic variants of FcRs are correlated with susceptibility to infectious diseases and cancers as well as the extent of clinical benefits from therapeutic antibodies against pathogens or malignancy. Indeed, examples of Fc engineering efforts that altered FcRs’ affinity showed improved clinical efficacy or prolonged half-life of antibodies [98,99,100,101,102,103,104,105]. This section introduces the variants of four different FcRs, which are FcγRIIa, FcγRIIIa, FcγRIIIb, and FcRn and affect the susceptibility to pathogens or cancer. This can be informative as to strategies for clinical application of antibody therapeutics.

### 3.1. FcγRIIa Genetic Variants: High-Affinity FcγRIIa for IgG2 Induces ADCP and Confers Low Susceptibility to Infection and Vaccine Effect

FcγRIIa is expressed on APCs such as DCs, macrophages, and B cells, and operates activating signaling pathway of effector functions by its own ITAM in the intracellular domain (Table 1) [22]. Recently, several studies suggested that FcγRIIa polymorphism affect the ‘vaccine effect’ through the FcγRIIa-mediated phagocytosis, which is part of the antigen presenting and hence can contribute to adaptive immune systems [106,107]. 

FcγRIIa polymorphism exists in the 131 residue of the C’E loop, which directly contacts with the Fc domain of IgG antibodies and results in the affinity difference (Figure 3 and Table 2) [108,109,110,111]. H131R polymorphism is amino acid sequence change from histidine (H; codon CAT) to arginine (R; codon CGT), and FcγRIIa-H131 has a higher affinity to the Fc domain than FcγRIIa-R131 (Figure 3). A notable feature of the FcγRIIa-H131 is its efficient binding with IgG2 as opposed to low or non-binding with the FcγRIIa-R131 [27]. Under bacterial infections, IgG2 subclasses are induced by bacterial capsular polysaccharide, which provokes ADCP by the interaction of IgG2-opsonized bacteria and the FcγRIIa-H131 [26]. For this reason, homozygous FcγRIIa-H/H131 patients showed lower susceptibility to bacterial infection than patients with other genotypes. Indeed, among pediatric patients who suffer from pneumococcal sepsis, the proportion of the R/R genotype (43%) was higher than the H/H genotype (22%) [112,113,114,115]. 

Ravetch and his colleagues demonstrated the feasibility of utilizing the FcγRIIa-mediated vaccine effect for both infection and cancer therapy in which the strengthened FcγRIIa affinity resulted in DCs maturation and induction of CD8+ T-cell response. Anti-influenza mAbs engineered to increase FcγRIIa affinity showed enhanced pharmacokinetic activity both in the prevention and treatment of influenza infection. In addition, the vaccine effect induced by FcγRIIa-expressing DCs was shown to lead to long-term anti-cancer activity [106,107].

### 3.2. FcγRIIIa Genetic Variants: High-Affinity FcγRIIIa for IgG Is Associated with Low Susceptibility to Cancer and Results in a Higher Response of NK Cell-Mediated ADCC of Therapeutic Antibodies

FcγRIIIa is expressed on natural killer (NK) cells, γδT cells, and subsets of monocytes and macrophages (Table 1) [116]. Once FcγRIIIa interacts with its ligand, ITAM in its cytoplasmic domain initiates the signaling pathway via phosphorylation by Src family kinases (SFKs) and spleen tyrosine kinases, thereby activating the effector function of the immune cells themselves, producing cytokines or chemokines, and inducing cell migration [117]. The effector function induced by FcγRIIIa includes ADCC, which is triggered by the interaction between multiple Fcs in ICs and the FcγRIIIa expressed on NK cells [118].

Two polymorphisms of FcγRIIIa have been identified in each extracellular domain 1 (EC1) and EC2 (Figure 3 and Table 2). In EC1, there is a tri-allelic polymorphism in which the residue 48 of the EC1 can be leucine (L; codon CTC), arginine (R; codon CGC), or histidine (H; codon CAC) [119]. In EC2, the 158 residue can be either valine (V; codon GTT) or phenylalanine (F; codon TTT). As the 158 residue directly contacts with the lower hinge region of IgG, this polymorphism affects the binding affinity to IgG (Figure 3) [120]. The V158 allele exhibits approximately twofold higher affinity for IgG than the F158 allele, which makes the NK cells more ready to induce ADCC [27]. Statistical analysis showed that carriers of the FcγRIIIa F158 allele appear to be more susceptible to cancer incidence. Indeed, among the patients with ERBB2/HER2-positive breast cancer, patients with the F/F158 genotype (46%) and F/V158 genotype (42%) have greater prevalence relative to those with the V/V158 genotype (12%). This proportion is consistent in the metastatic colorectal cancer patients: the F/F158 genotype (48.5%) and F/V158 genotype (41.4%) were observed for a higher proportion than the V/V158 genotype (10.1%) [35,36]. Notably, in addition to the statistics that FcγRIIIa polymorphism affects cancer prognosis, patients with V/V158 genotype also display more clinical benefits than those with F/F158 and F/V158 genotype in the treatment of therapeutic antibodies, including rituximab (Rituxan^®^), trastuzumab (Herceptin^®^), cetuximab (Erbitux^®^), and avelumab (Bavencio^®^) [35,121,122,123]. These results are presumed to be resulting from higher responses of NK cell-mediated ADCC due to the higher affinity to mAbs in V158 allele carriers compared with people with the F/F158 homozygote [124]. 

To improve the efficacy of cancer immunotherapy with regards to NK cell-mediated ADCC, increasing the FcγRIIIa affinity to the antibody Fc can be a powerful strategy. Fc-engineered anti-cytotoxic T-lymphocyte-associated protein 4 (CTLA-4) antibody, AGEN1181, enhanced its binding to both F158 and V158 alleles, thus showing improved therapeutic efficacy in patients who carry the F158 allele, compared with the first generation anti-CTLA-4 antibody treatment [125]. Therefore, patients with low affinity FcγRIIIa allele, the remaining 40% of patients who had insignificant treatment effects with the first generation anti-CTLA-4 antibody, would be able to receive clinical benefits from the Fc-engineered antibodies.

### 3.3. FcγRIIIb Genetic Variants: High-Affinity FcγRIIIb for IgG1 and IgG3 Induces Neutrophil-Mediated Phagocytosis and Contributes to Low Susceptibility to Infections

FcγRIIIb is an IgG receptor with a glycosylphosphatidylinositol anchor in the transmembrane domain of neutrophils, basophils, and eosinophils (Table 1) [126]. The activation of FcγRIIIb leads to neutrophil degranulation and oxidative burst and induces the phagocytosis of IgG-opsonized pathogens by neutrophils [127].

FcγRIIIb has three polymorphisms in the membrane-distal Ig-like extracellular domain, each of which is a haplotype, comprising five respective amino acid residues: NA1 (R36, N65, A78, D82, and V106), NA2 (S36, S65, A78, N82, and I106), and SH (S36, S65, D78, N82, and I106) (Figure 3 and Table 1) [128]. The two co-dominantly expressed allelic variants, NA1 and NA2, show the difference in four of the five amino acids, which changes the number of glycosylation sites: NA1 has four and NA2 has six, which affects their interactions with IgG (Figure 3) [129]. The NA1 allele has a higher affinity for IgG1 and IgG3, and this allows the NA1 allele to more efficiently induce the neutrophil-mediated phagocytosis of IgG-opsonized particles and bind IgG3 more effectively than the NA2 allele [130]. The polymorphism of FcγRIIIb also affects the susceptibility to immune disorder or malaria infection. The risk of developing immune-mediated complications for chronic granulomatous disease (CGD) was lower in NA1 homozygous patients, compared with those with NA1/NA2 or NA2/NA2 [131]. Severe malaria infection in children was shown more in people carrying the FcγRIIIb-NA2 allele than those with NA1: patients with NA2/NA2 were 51.6%; NA1/NA2, 38.5%; and NA1/NA1, 9.9% [132].

Increasing FcγRIIIb affinity is expected to improve anti-cancer efficacy through neutrophil-mediated phagocytosis. Obinutuzumab (Gazyva^®^) is a type 2 anti-CD20 antibody approved for chronic lymphocytic leukemia (CLL). Obinutuzumab, the glycoengineered anti-CD20 for enhanced binding to FcγRIIIb showed a seven-fold higher affinity relative to the wild-type rituximab, which results in 47% higher phagocytosis of opsonized CLL by activating the FcγRIIIb-expressing polymorphonuclear neutrophils (PMNs) [133].

### 3.4. FcRn Genetic Variants: High-Expression of FcRn Increases IgG Half-Life

Heterodimeric FcRn is a family of major histocompatibility complex (MHC) class I molecule, which comprises the α-chain and β2m. It is expressed in monocytes, macrophages, neutrophils, DCs, and on the surface of epithelial and endothelial cells (Table 1). The FcRn interacts with both IgG and albumin at the acidic environment (pH ≤ 6.5) to be sorted into vesicles so as to be released back into the bloodstream without being decomposed in the lysosome. Consequently, the sorted vesicles are released out to the serum at a neutral pH (pH 7.4), which is how FcRn contributes to an important role in maintaining the high amount of IgG and albumin in circulation [24].

The expression level of FcRn depends upon a variable number of tandem repeats (VNTR) polymorphisms (Figure 4 and Table 2). VNTR identified within the promoter of the *FCGRT* gene may influence the transcriptional activity of the *FCGRT* gene promoter by changing the length of the transcription regulatory region, thereby altering the expression level of FcRn and affecting IgG catabolism (Figure 4) [134,135]. The VNTR region in the promoter consisted of one to five tandem repeats (VNTR1–VNTR5). VNTR3, the most common allele, is associated with an increase in promoter activity and the FcRn protein level, compared with the VNTR2 allele. In practice, monocytes from VNTR3 homozygotes displayed increased FcRn expression and led to recycling IgG more efficiently than the VNTR2/VNTR3 heterozygotes [134]. The IgG replacement of common variable immunodeficiency disorder (CVID) patients treated with intravenous immunoglobulin (IVIg) was more efficacious in VNTR3 homozygous patients than in VNTR2/VNTR3 heterozygous patients. Therefore, high IgG efficiency was found in CVID patients with VNTR3 homozygotes [136,137]. 

Since the expression level of FcRn depends on the VNTR variants and it is associated with IgG half-life in blood, the modulation of FcRn-mediated IgG recycling can be applied to various diseases with two opposite strategies. The first is to increase the half-life of therapeutic antibodies for infection and cancer treatment, thus expecting the advantage of less frequent and lower doses of therapeutic antibody administration. Fc variants with M428L/N434S mutations showed an 11-fold improvement in FcRn binding affinity at pH 6.0 and a 3-fold improvement in serum half-life than native IgG1 in cynomologus monkeys [138]. Furthermore, Georgiou and his colleagues developed an IgG1 Fc variant containing the L309D/Q311H/N434S substitutions that bind strongly at the acidic pH and simultaneously are released from FcRn at neutral pH showed improved pharmacokinetics relative to both endogenous IgG1 and widely used half-life extension variants [99]. The second is to reduce circulating IgG by blocking FcRn-IgG interaction for autoimmune disease treatment by decreasing the presence of endogenous pathogenic autoantibodies [139]. Rozanolixizumab, an anti-FcRn antibody, lasted dose-dependent reduction in serum IgG concentrations [140]. Additionally, in a phase 2 clinical trial, the clinical efficacy and safety of rozanolixizumab were confirmed in the moderate-to-severe generalized myasthenia gravis (gMG) patients, and phase 3 is ongoing [141]. Another anti-FcRn antibody, HBM9161, resulted in sustained and dose-dependent IgG reduction in a phase 1 clinical trial [142].

## 4. Conclusions

We classified the mutation positions of FcR variants into three groups in this article. Firstly, those in the ectodomain of FcγRs directly affect the interaction between FcRs and antibody Fc, which can differ in the extent of the recruitment of immune effector leukocytes such as NK cells for ADCC activity or macrophages for ADCP activity. This interaction can be strengthened by antibody Fc engineering with enhanced affinity to distinct FcγRs. Secondly, polymorphisms in the transmembrane or cytoplasmic domain of FcεRs or FcαRs on mast cells or PMNs, respectively, determine whether the downstream intracellular signaling would be turned on or off. For example, the G237 allele in the FcεR β subunit of cytoplasmic tail in proximity to ITAM motif on γ subunit is more easily activated and leads to hypersensitive immune response, relative to the E237 allele. Lastly, FcRs variants in the transcription regulatory region can affect the expression of the respective genes, which can change the activation threshold of immune effector functions [31,45,143].

In the aspect of disease susceptibility, the enhancement of not only FcRs’ affinities to Igs but also FcR-mediated signal transduction, as well as FcRs transcription via FcRs variants, would increase the risk of autoimmune diseases. Conversely, lowering the affinities, downstream signaling, or transcription level of FcRs weaken the protectiveness against the pathogenesis of infections or outbreaks of malignancy. Especially, the Fc–FcRs affinities can be therapeutically surmountable through Fc engineering for either boosting or attenuating the interaction. However, the pharmacologic activities of therapeutic antibodies that harnesses the effector functions is dependent upon patient genotype [144,145,146]. Attempts to increase effector functions by engineered Fc have shown a remarkably improved therapeutic potency in low-affinity receptor genotype patients. For example, Margetuximab (Margenza^®^) was designed to target the same epitope with trastuzumab (Herceptin^®^) with enhanced affinity to FcγRIIIa. In a SOPHIA phase 3 clinical trial, the FcγRIIIa-V/F158 and FcγRIIIa-F/F158 genotype patients showed an improved therapeutic potency of HER-2 positive breast cancer but did not show a significant effect in the *v*/*v* genotype patients [147,148]. This result showed that the treatment effect can differ depending on the FcRs’ genotype. Therefore, the clinical benefits of engineered antibodies can be maximized only when the genotypes of FcRs in patients are well-studied and considered in the treatment of mAbs with a proper therapeutic window for each individual.

## Figures and Tables

**Figure 1 ijms-22-09489-f001:**
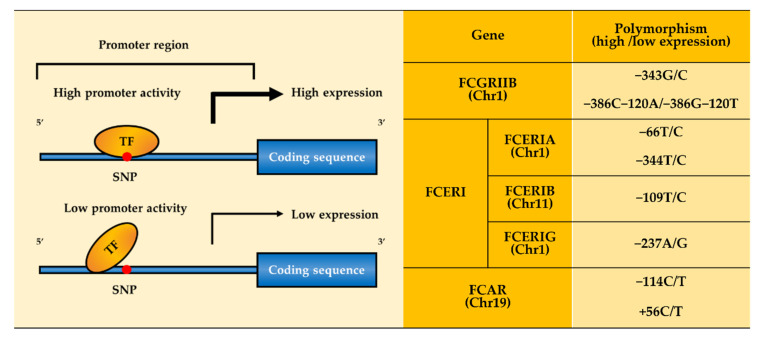
Polymorphisms influence gene expression. Polymorphism in the promoter region induces differences in promoter activity and therefore changes the expression level of Fc receptors. −386C−120A and −343G of FCGRIIB increase the expression level of FcγRIIb. −66T and −344T of FCERIA, −109T of FCERIB, and −237A of FCERIG increase the expression level of α, β, and γ subunit, respectively. −114C and +56C of FCAR increase the expression level of FcαRI.

**Figure 2 ijms-22-09489-f002:**
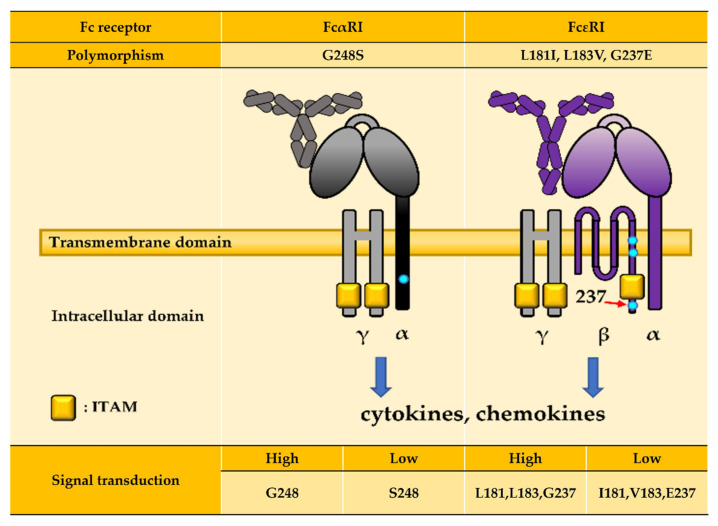
Polymorphisms alter signal transduction. Turquoise dots indicate amino acid sequences that affect downstream signaling pathways. G248 located in the cytoplasmic domain of FcαRI, produces pro-inflammatory cytokines without γ-chain, compared to S248. L181 and L183, located in the transmembrane domain of FcεRI, are more easily seen in patients with atopy and asthma, compared to I181 and V183, respectively. G237, located in the intracellular domain of FcεRI, causes an excessive allergic response compared to E237.

**Figure 3 ijms-22-09489-f003:**
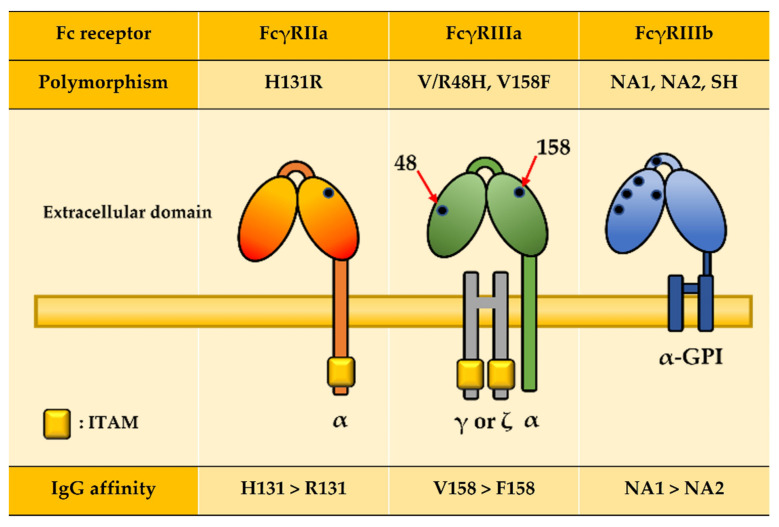
Polymorphisms affect immunoglobulin affinity. Black dots indicate amino acid residues that affect IgG affinity. H131 of FcγRIIa, V158 of FcγRIIIa, and NA1 of FcγRIIIb havehigher IgG affinity than the other alleles.

**Figure 4 ijms-22-09489-f004:**
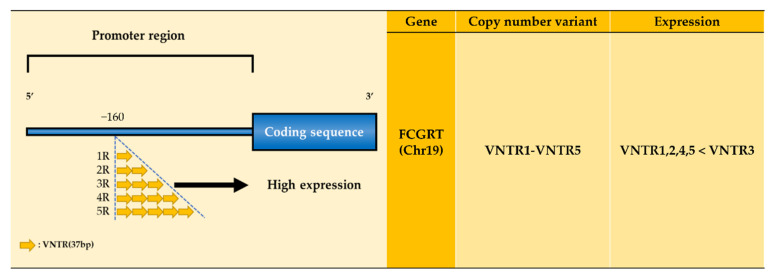
Repeat 3 of VNTR in the promoter of the FCGRT gene shows the highest FcRn expression.

**Table 1 ijms-22-09489-t001:** Ig binding affinity, function, and expression of individual FcRs.

Receptor	Ig Isotype and Subclass Binding Affinity		Function	Expression		
				Myeloid Cell	Lymphoid Cell	Non-Immune Cell
FcγRI	IgG1,2,3,4	High affinity (IgG1 = IgG3 > IgG4 >>> IgG2)	Activation	Monocyte, Macrophage, Neutrophil, DC, Mast cell, Eosinophil	-	-
FcγRIIa		Low affinity (H131: IgG3 > IgG1 > IgG2 > IgG4 R131: IgG3 > IgG1 > IgG4 > IgG2)	Activation/Inhibition	Monocyte, Macrophage, Neutrophil, DC, Mast cell, Basophil, Eosinophil	T cell	Platelet, Endothelium
FcγRIIb		Low affinity (IgG3 > IgG1 > IgG4 >> IgG2)	Inhibition	Monocyte, Macrophage, Neutrophil, DC, Mast cell, Basophil, Eosinophil	B cell, Plasma cell	-
FcγRIIc		Low affinity (IgG3 > IgG1 > IgG4 >> IgG2)	Activation	Monocyte, Macrophage, Neutrophil	NK cell	-
FcγRIIIa		Low affinity (IgG3 >> IgG1 > IgG4 > IgG2)	Activation/Inhibition	Monocyte, Macrophage	NK cell, T cell	-
FcγRIIIb		Low affinity (IgG1 = IgG3 >>> IgG2 = IgG4)	Activation	Neutrophil, Basophil, Eosinophil	-	-
FcRn		High affinity (pH ≤ 6.5)/Low affinity (pH 7.4) (IgG4 > IgG1 > IgG3 > IgG2)	Transcytosis/Recycling	Monocyte, Macrophage, Neutrophil, DC	-	Endothelium, Epithelium
FcεRI	IgE	High affinity	Activation	Langerhans cell, Eosinophil, Mast cell, Basophil, DC, Monocyte	-	Platelet
FcεRII		Low affinity	Activation	Langerhans cell, Monocyte, DC, Macrophage, Eosinophil,	B cell, T cell, NK cell	Platelet
FcαRI	IgA1,2	Low affinity (IgA1 = IgA2)	Activation/Inhibition	Monocyte, Macrophage, Neutrophil, DC, Kupffer cell	-	-
FcμRI	IgM	Low affinity (monomeric IgM)/High avidity (pentameric IgM)	NA	-	B cell, T cell, NK cell	-
Fcα/μR	IgA/IgM	High affinity (IgM > IgA)	NA	Macrophage, DC	B cell	-
FcδR	IgD	NA				
plgR	Dimeric IgA/pentameric IgM	High affinity	Transcytosis	-	-	Endothelium, Epithelium

DC: dendritic cells, NK cells: natural killer cells, NA: not analyzable.

**Table 2 ijms-22-09489-t002:** Summary of genetic variants of individual FcRs and related diseases.

Fc Receptor	Genetic Variant						Related Diseases
	Polymorphism					Copy Number Variant	
	Promoter Region		Extracellular Domain	Transmembrane Domain	Intracellular Domain		
FcγRIIa			H131R				R131 allele: Bacterial infection (Low affinity to IgG2)
FcγRIIb	−120T/A −343G/C −386G/C						−343G, −386C−120A allele: SLE (High FcγRIIb expression)
FcγRIIIa			48L/R/H V158F				F131 allele: Cancer (Low affinity to IgG)
FcγRIIIb			NA1 NA2 SH				NA2 allele: CGD (Low affinity to IgG1 and IgG3)
FcRn						VNTR1- VNTR5	NA
FcεRI	α-subunit	−66T/C −344C/T					−66T, −344T allele: Allergic diseases (High FcεRI expression)
	β-subunit	−109C/T		I181L V183L	E237G		−109T allele: Asthma (High FcεRI expression) L181, L183 allele: Atopy, Asthma (Altered signaling pathway leading to cell activation) G237 allele: Allergy diseases (Activating signal induction)
	γ-subunit	−237A/G					−237A allele: AIA (High FcεRI expression)
FcαRI	−114C/T +56C/T				S248G		−114C, +56C allele: IgAN (High FcαRI expression) G248 allele: SLE (Altered signaling pathway leading to cell activation)

NA: not analyzable.

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
