# Peer review of "Fc Receptor Variants and Disease: A Crucial Factor to Consider in the Antibody Therapeutics in Clinic"

_ijms, 2021, doi:10.3390/ijms22179489_

Round 1
Reviewer 1 Report
Very well summarized review of Fc receptor variants and disease.
This paper is a good summary of Fc and FcR interactions and it’s role in disease. The authors have touched upon important mechanisms in this context.Author Response
We wish to thank you for your encouraging remarks. We have further revised the manuscript using the “Track Changes” function in MS Word file.
Reviewer 2 Report
The authors write a comprehensive review on Fc mediated immune effector functions in relation to FcR genetic diversity. The authors propose how engineering of Fc portions in antibody-based therapy may shape the outcome of treatment, as well as mention the impact of the host’s genetic background (e.g. FcR variants) on the response. The authors conclude the latter to be a critical aspect to take into consideration to predict and analyze the clinical benefits of Ab therapy. They suggest engineering as means of improving therapeutic efficiency but in the context of personalized medicine.
Comments:
1. Within the citations are included other review articles. Ideally, original papers should be cited.
2. The English language requires revisions, of which I only bring here a few examples:
line 114 Single nucleotide polymorphisms (SNPs) in the promoter region of the FCGRIIB gene……. Three SNPs in the promoter
lines 192-195 is very difficult to read, please re-write.
line 235 The FCAR gene promoter with these two alleles have a significantly higher ( use has or if preferable use promoters)
line 264 this can be informative as to strategies for clinical application…
Author Response
The authors write a comprehensive review on Fc mediated immune effector functions in relation to FcR genetic diversity. The authors propose how engineering of Fc portions in antibody-based therapy may shape the outcome of treatment, as well as mention the impact of the host’s genetic background (e.g. FcR variants) on the response. The authors conclude the latter to be a critical aspect to take into consideration to predict and analyze the clinical benefits of Ab therapy. They suggest engineering as means of improving therapeutic efficiency but in the context of personalized medicine.
: Thank you for your encouraging remarks and helpful comments. We have addressed your comments in detail below and further revised the manuscript using the “Track Changes” function in MS Word file.
Comments:
- Within the citations are included other review articles. Ideally, original papers should be cited.
: As recommended, we have re-cited the references #8, #21, #22, #24, and #25 of our manuscript.
- The English language requires revisions, of which I only bring here a few examples:
: In addition to the kindly suggested comments, we have further proof-read the manuscript, using the “track-change” function.
line 114 Single nucleotide polymorphisms (SNPs) in the promoter region of the FCGRIIB gene……. Three SNPs in the promoter
: On the basis of your kind recommendations, we have modified the line 115 in the revised manuscript our manuscript.
lines 192-195 is very difficult to read, please re-write.
: As recommended, we have re-written the lines 190-201 in the revised manuscript to be more readable.
G237E polymorphism may also regulate the cell surface expression by the formation of the two different types of β subunits, which are a full-length and a truncated β chain. Kinet et al . identified a new transcript of β subunit via an in-frame stop codon generated by alternative splicing of FcεRI β transcript and translated into truncated β subunit [74]. Unlike the full-length β subunit, the truncated one reduces the cell surface expression of FcεRI and the G237 residue, which is close to the splicing site might affect splicing that leads to the high level of the full-length β chain, which can induces more functional IgE receptor and higher allergic responses [74-76]
- G237E polymorphism, which is close to an alternative splicing site may also affect expression level of the receptor on the cell surface. Kinet and his colleagues identified a new transcript of the β subunit that contains an in-frame stop codon generated by an alternative splicing of the FcεRI β transcript, which would be translated into a truncated β subunit [74]. Compared to the full-length β subunit, the truncated one is significantly less expressed and therefore the G237E polymorphism might affect the expression level of the β chain, which can lead to the diverse susceptibility of allergic responses among individuals [74-76].
line 235 The FCAR gene promoter with these two alleles have a significantly higher ( use has or if preferable use promoters)
: On the basis of your kind recommendations, we have modified the line 241 in the revised manuscript.
line 264 this can be informative as to strategies for clinical application…
As recommended, we have modified the wrong format in the line 271 in the revised manuscript.